# Leukocyte telomere length as a compensatory mechanism in vitamin D metabolism

**Deniz Agirbasli**[1,2]*, **Minenur Kalyoncu**[3], **Meltem Muftuoglu**[3,4], **Fehime Benli Aksungar**[5,6], **Mehmet Agirbasli**[7]

**1** Department of Medical Genetics, Cerrahpaşa Faculty of Medicine, Istanbul University-Cerrahpaşa, Istanbul, Turkey, **2** Department of Medical Biology, School of Medicine, Mehmet Ali Aydinlar University, Istanbul, Turkey, **3** Department of Medical Biotechnology, Institute of Health Sciences, Acibadem Mehmet Ali Aydinlar University, Istanbul, Turkey, **4** Department of Molecular Biology and Genetics, Acibadem Mehmet Ali Aydinlar University, Istanbul, Turkey, **5** Department of Biochemistry, School of Medicine, Mehmet Ali Aydinlar University, Istanbul, Turkey, **6** Acıbadem Labmed Clinical Laboratories, Atasehir, Istanbul, Turkey, **7** Department of Cardiology, T.C Istanbul Medeniyet University, School of Medicine, Ministry of Health, Goztepe Prof. Dr. Suleyman Yalcin City Hospital, Istanbul, Turkey

* deniz.agirbasli@iuc.edu.tr

**Data Availability Statement:** All relevant data are within the manuscript and its Supporting Information files.

## Abstract

Vitamin D deficiency is common among postmenopausal women. Telomere length can be a potential protective mechanism for age-related diseases. The objective of our study is to examine the association of vitamin D supplementation on leukocyte telomere length (LTL) in healthy postmenopausal women with vitamin D deficiency. The study was designed as a placebo-controlled study to investigate the short-term effects of vitamin D supplementation and seasonal changes on vitamin D related parameters, including 25(OH)D, 1,25(OH)$_2$D parathormone (PTH), Vitamin D binding protein (VDBP), vitamin D receptor (VDR), and telomere length in a cohort of postmenopausal women (n = 102). The group was divided as supplementation (n = 52) and placebo groups (n = 50). All parameters were measured before and after treatment. Serum VDBP levels were measured by ELISA method and *VDR*, *GC (VDBP)* gene expressions and relative telomere lengths were measured in peripheral blood mononuclear cells (PBMC) using a quantitative real-time PCR method. The results demonstrate that baseline levels were similar between the groups. After vitamin D supplementation 25(OH)D, 1,25(OH)$_2$D, PTH and VDBP levels were changed significantly compared to the placebo group. At the end of the study period, LTL levels were significantly increased in both groups and this change was more prominent in placebo group. The change in *GC* expression was significant between treatment and placebo groups but VDR expression remained unchanged. Even though the study was designed to solely assess the effects of vitamin D supplementation, LTL was significantly increased in the whole study group in summer months suggesting that LTL levels are affected by sun exposure and seasonal changes rather than supplementation. The study displayed the short-term effect of Vitamin D supplementation on vitamin D, PTH levels, LTL and vitamin D associated gene expressions. The relation between Vitamin D and LTL is not linear and could be confounded by several factors such as the population differences, regional and seasonal changes in sun exposure.

**Funding:** The study is funded by The Scientific and Technological Research Council of Turkey (TUBITAK) 3001 grant with the project number 116Z812 DA is the principle investigator. https://www.tubitak.gov.tr The funders had no role in study design, data collection and analysis, decision to publish, or preparation of the manuscript.

**Competing interests:** The authors have declared that no competing interests exist.

## Introduction

Vitamin D is a steroid hormone synthesized in the skin via sunlight. It plays an important role in biological systems including mineralization of bones, stabilization of blood calcium levels, modulating the innate and adaptive immune responses, and nerve conduction. Vitamin D deficiency is defined as 25-hydroxy-vitamin D (25(OH)D) concentrations lower than 20 ng/ml (50 nmol/l) whereas insufficiency is defined as levels ranging from 20 to 29.9 ng/ml [1]. Population studies indicate that 25(OH)D deficiency is a risk factor for common chronic complex diseases such as cardiovascular disease, diabetes mellitus, neuropsychiatric disorders, and autoimmune diseases. Obesity and vitamin D deficiency in women are important public health problems [2, 3]. Vitamin D deficiency can be seen at any age and is highly prevalent in postmenopausal women due to the decrease in estrogen levels [4]. Aging, insufficient exposure to sunlight, obesity, diet, and hyperlipidemia are also risk factors for vitamin D deficiency. For instance, among the elderly population, 61% in the United States, 90% in Turkey, 96% in India, 72% in Pakistan, and 67% in Iran are vitamin D deficient [5].

Although there is still controversy about the amount and duration of vitamin D supplementation, vitamin D deficient adults require 6000 IU/day of vitamin $D_3$ for 8 weeks or 50,000 IU of vitamin $D_3$ once weekly for 8 weeks [1]. Seasonal variations affect vitamin D levels but as a carrier protein for vitamin D metabolites; vitamin D binding protein (VDBP) levels remain mostly stable [6].

Telomere length is an important parameter in chromosome stability and chronic diseases. Telomeres are repetitive DNA sequences at the ends of the chromosomes and protect the chromosomal integrity. Telomeres maintain the genomic and cellular stability. Telomere length decreases with aging, inflammation and oxidative stress. Telomere length shortening is associated with many chronic diseases [7, 8].

Vitamin D is known to be protective for aging and age-related chronic diseases. It also plays a role in the cell's vital activities such as differentiation, proliferation and apoptosis [9]. Vitamin D levels are related to aging and telomere length as it reduces inflammation and is related to genomic stability [10, 11]. 1,25- Dihydroxyvitamin D (1,25(OH)$_2$D) receptors are present in leukocytes and this may support the effect of vitamin D in leukocyte telomere length (LTL) [12]. There are numerous studies investigating the LTL in postmenopausal women. Although ethnicity does not play a role in LTL, age and gender are among the determinants of LTL [13, 14]. Decreased estrogen levels after menopause, a pivotal factor in the biology of aging, were positively associated with LTL [13]. Studies also indicate from the premenopausal period through the perimenopausal period to the postmenopausal period there is gradual attrition in LTL which then turned out to be more stable after the postmenopausal period [15].

Previous studies demonstrated the association between telomere length and vitamin D levels in cross sectional studies [16]; however, the short-term effects of vitamin D supplementation on telomere length remain to be elucidated. The effects of Vitamin D supplementation or seasonal changes on telomere length remain largely unknown. Therefore, this study was designed as a placebo-controlled study to investigate the short-term effects of vitamin D supplementation on vitamin D related parameters, including 25(OH)D, 1,25(OH)$_2$D and PTH, VDBP, VDR, and telomere length in a cohort of postmenopausal women.

## Materials and methods

### The study population

Healthy postmenopausal women with serum vitamin D levels < 20 ng/ml (<50 nmol/l) (n = 102) were included in the study. According to the definition of the World Health

Organization (WHO); health is a state of complete physical, mental, and social well-being. The study population was defined as healthy, indicating that they are independent of any known diseases without any prior discerning medical history in accordance with the definition of WHO. The participants were chosen from the volunteers who have not been exposed to vitamin D supplements for at least one year prior to the study. Menopause was defined as the cessation of menstrual periods for twelve consecutive months [17].

The participants were all chosen from the primary health care center in Manyas, Balikesir; a city located in the south Marmara region of Turkey. We selected all participants from the same region to avoid differences in sunlight exposure. Subjects were given case numbers and identities were kept confidential. Information on height, weight, dietary habits, physical activity, alcohol intake, and smoking habits was collected by self-administered questionnaires. All patients underwent physical examination. Body mass index (BMI) was calculated as weight (kg)/height (m$^2$). Cigarette smoking status was defined as the consumption of 100 cigarettes per lifetime for a current cigarette smoker. Never-smoker was defined as consumption of less than 100 cigarettes per lifetime. Subjects were divided into three groups by self-reports for the level of physical activity as non-exercisers, mild exercisers (<4 hours/week), moderate to high exercisers (> 4 hours/week). Physical activity was defined as the performance of any structured activity on a regular basis such as walking, lifting weights, doing aerobics, resistance training, or riding a stationary bike [18]. Subjects with chronic diseases (renal insufficiency, megaloblastic anemia, cardiovascular disease, history of cancer, cardiovascular disease, thyroid or parathyroid disease, liver dysfunction, kidney disease, extensive skin disease, extreme stress / depression, malabsorption / malnutrition, chronic inflammatory disease, hyperlipidemia requiring statin therapy) and subjects using medications that alter serum vitamin D levels within one year, carriers of mutations that reduces vitamin D levels and causing hereditary diseases, women who follow a strict vegan diet were excluded from the study. The study was conducted according to the principles expressed in the Declaration of Helsinki. Acıbadem University Institutional Review Board (ATADEK) approved the study protocol with the decision number 2016-1/17. Written informed consent was obtained from all participants. When the type 1 error is set as 5% with study power 80%, with a hypothetical difference of 10% between the telomere lengths of the study groups, the minimum sample size is calculated as 38 for each group.

## Vitamin D supplementation

Serum vitamin D (25(OH)D) levels of all subjects were measured at the beginning of the study. The subjects who were eligible to participate in the study (vitamin D levels < 20 ng/ml), (n = 102) were randomly selected. Randomization was performed by toss of a coin method. The treatment group (n = 52) received oral vitamin D$_3$ supplementation (cholecalciferol) at a dose of 50,000 IU/week for 8 weeks (Devit-3$^®$ Deva oral solution, Turkey) and the placebo group (n = 50) received sunflower oil as a placebo of which participants consumed 15 ml once a week for 8 weeks.

Sunflower oil was chosen as placebo, since prior studies stated that it did not have a significant effect on vitamin D levels [19–21]. Subjects and the laboratory staff were blinded to the study arm. Vitamin D supplementation bottles were packaged to look identical to ensure that the participants were truly blinded. After the study, the placebo group received vitamin D supplements free of charge at the same amount given to the treatment group. Since 25(OH)D had a long half-life and was the predominant form in the circulation, it was used as the primary marker of vitamin D status [22].

To detect the effect of seasonal variations on vitamin D supplementation, we analyzed LTL according to the season of treatment. The summer group was the participants taking vitamin

D or placebo starting in May-June; the winter group was the participants taking vitamin D or placebo starting in October-January. After 2 months of treatment and 1 month waiting period, peripheral blood was taken and the parameters were re-measured.

## Laboratory assessment

**Blood samples and peripheral blood mononuclear cell (PBMC) isolation.** Peripheral blood samples were drawn at 9 a.m. Blood samples were taken into BD Vacutainer® SST™ Tubes (8 cc) for vitamin D measurements and ELISA analysis; and purple-cap EDTA tubes (10 cc) for genetic measurements. Serum was obtained by centrifugation for 6 minutes at 3000 rpm, +4˚C within 1 hour after the blood collection. Serum was aliquoted into 1.5 ml Eppendorf tubes and stored at -80˚C for ELISA and vitamin D measurements.

PBMCs were isolated from peripheral blood using the Ficoll-Paque gradient centrifugation method within 6 hours after blood collection [23]. PBMCs were used for DNA and RNA isolation.

**Biochemical analysis.** All biochemical measurements were carried out in ISO-15189 accredited Acibadem Labmed Clinical Laboratories, Istanbul, Turkey. Biochemical measurements of 25(OH)D levels were carried out by CLIA method on ADVIA Centaur® XPT Immunoassay System, Siemens, Germany. Briefly, serum samples in eppendorf tubes were centrifuged at 1000 g for 10 minutes before being placed in the instrument, transferred to cuvettes and incubated for 15 seconds. After incubation with 200 μl reactive auxiliary pack reagent at 37˚C for 4.5 minutes, with 50 μl lite reagent at 37˚C for 5.5 minutes, with 100 μl solid phase reagent and with 50 μl auxiliary well reagent at 37˚C for 3 minutes, aspiration of the reagent by separating the solid phase from the mixture was performed by the device. The chemiluminescence reaction was initiated and the results were reported. Intra-assay and inter-assay CV of the measurements were 2.29 and 5.16 respectively.

PTH serum levels were measured by the ADVIA Centaur PTH test which is a direct method using fixed amounts of 2 anti-human PTH antibodies. It is a sandwich immunoassay using chemiluminometric technology. The first (N-terminal) antibody in Lite Reagent is a monoclonal mouse anti-human PTH labeled with acridinium ester. The second antibody is a biotinylated monoclonal mouse anti-human PTH (C-terminal) antibody bound to Streptavidin coated paramagnetic latex particles in Solid Phase. The reference values for PTH levels were 18.5–88 pg/ml. Serum $1,25(OH)_2D$ levels were measured by IDS-iSYS 1,25-Dihydroxy Vitamin D assay involving immunopurification followed by quantitative determination of serum $1,25(OH)_2D$ on IDS-iSYS system (IDs-iSYS Multi-Discipline Automated System, Immunodiagnostic Systems Limited, Tyne & Wear, UK). Intra-assay and inter-assay CV of the measurements were 3.76 and 7.59 respectively. The reference values for $1,25(OH)_2D$ were 26.1–95.0 pg/ml.

**Measurement of leukocyte telomere length.** DNA was isolated from mononuclear cells using Quick-DNA™ Miniprep Plus Kit (Zymo Research®, Irvine CA, USA). The purity and concentration of DNA samples were measured by NanoDrop™ 2000 /2000c spectrophotometer (Thermo Scientific Inc). DNA samples with A260/280 ratio between 1.8–2 nm and A260/230 ratio between 2–2.2 nm were selected and stored at -20˚C for the measurement of telomere length. The quantitative PCR (qPCR) method was performed for the determination of LTL [24]. The primer sequences for the amplification of telomeres (T), single copy gene 36B4 (S), and β-globin were described by Cawthon, 2002. QPCR was performed on the BIORAD CFX96 Touch™ Real Time PCR detection system using the SensiFAST™ SYBR® No-ROX Kit (Bioline Meridian, USA). The amount of DNA for qPCR was calculated from the standard curve obtained from the dilution series as described [24]. Each sample was studied in duplicate.

Telomere length, expressed as telomere to single-copy gene ratio (T/S), was assessed by qPCR. The ΔCt values were calculated for each subject. ($\Delta Ct = averageCt36B4 - average\ Ct_{telomere}$).

**cDNA synthesis and vitamin D receptor (*VDR*), vitamin D-binding protein gene (*GC/VDBP*) expressions.** RNA was isolated from mononuclear cells using Quick RNA Miniprep Plus Kit (Zymo Research®, Irvine CA, USA). The purity and concentration of DNA samples were measured by NanoDrop ™ 2000/2000c spectrophotometer (Thermo Scientific Inc). RNA samples with A260/280 ratio between 1.9–2.2 nm were selected and stored at -80°C for *VDR* and *GC* mRNA expression studies. In order to determine *GC* and *VDR* expression, the same amount of RNA was taken from each sample and cDNA synthesis was performed. cDNA synthesis was performed with SensiFAST cDNA synthesis kit (Zymo Research®, Irvine CA, USA). Quantitative real-time PCR (qRT-PCR) was performed using DNA specific primers (at least one of the primer pairs was designed with exon-intron junction) (Table 1). Primers were designed by Primary BLAST software available at https://www.ncbi.nlm.nih.gov/tools/primer-blast/. B-Actin (*ACTB*) was used as an internal control. B-Actin primers were prepared as described [25]. Fluorescence detection of samples in qRT-PCR was performed on the BIORAD CFX96 Touch ™ Real-Time PCR detection system using Sensi FAST™ SYBR® No-ROX Kit (Bioline Meridian, USA). The amount of expression of the genes was determined by ΔΔCT method.

**Measurement of VDBP levels by ELISA.** VDBP serum levels were measured by a commercially available sandwich-type ELISA (ELISA VDBP, SEB810Hu, Cloud Clone, USA), according to the manufacturer's protocol. Briefly, standards and sample sera, respectively, were added to the wells on the plate coated with antibodies specific for VDBP. After incubation at room temperature, unbound material was washed off, secondary antibodies specific to the target protein, conjugated with horse radish peroxidase (HRP) was added. After incubation and addition of stop solution after washing, the plate was read at an appropriate wavelength (450 nm) on a spectrophotometer (Biotek, Vermont, USA). A standard curve was generated with standard absorbance values and the amount of target proteins in the subject samples was determined.

In the ELISA test, how close the absorbance values were to the standard values was calculated with the correlation coefficient predisposed to 0.95. Values above $R^2 = 0.9967$ showed that the positive standards of all samples were working properly.

## Statistics

All statistical analyzes were performed using SPSS program (version 20.0 for Windows, SPSS Inc. Chicago, IL). The Kolmogorov-Smirnov or Shapiro-Wilk test was used to analyze the normality of the data. Continuous variables with normal distribution were expressed as mean ± standard deviation, and variables with nonparametric distribution were expressed as median and interquartile range (IQR) and categorical variables were expressed as percentages

**Table 1. Designed primers for *VDR* and GC genes.** *ACTB* gene was used for internal control.

| Gene | Primer Sequence |
|---|---|
| *VDR* | F: 5′–CCAGGATTCAGAGACCTCACC–3′ |
| | R: 5′–AATCAGCTCCAGGCTGTGTC–3′ |
| *GC (VDBP)* | F: 5′–CAAGGCTCAGCAATCTCAT–3′ |
| | R: 5′–CTCTTTGGCCATGCAATC–3′ |
| *ACTB(B-ACTIN)* | F: 5′–GCACAGAGCCTCGCCTT–3′ |
| | R: 5′–GTTGTCGACGACGAGCG–3′ |

when appropriate. Student's t-test or Mann Whitney U test was used to compare unpaired samples as needed. The relationships among parameters were assessed using Pearson's or Spearman's correlation coefficient according to the normality of the data. A p-value of <0.05 was considered statistically significant.

## Results

### Demographic and biochemical results of the study group

This study included healthy 102 postmenopausal women (25(OH)D<20 ng/ml (<50 nmol/l)) who had never used vitamin D supplements or hadn't used vitamin supplements for at least a year. Subjects were divided into two groups as Vitamin D group taking vitamin D supplements (n = 52) and the placebo group (n = 50). Subjects were randomly selected for allocating vitamin D supplement/placebo by toss of a coin method.

Enrollment of subjects started in October 2017, ended in June 2018. Demographic variables were displayed in Table 2. Baseline characteristics were similar between the groups. The distribution between Vitamin D and placebo groups was similar for smoking and physical activity status (Table 2). Baseline biochemical parameters were distributed homogenously in both groups and were shown in Table 3.

Vitamin D supplementation and placebo groups have similar 25(OH)D levels before vitamin D treatment (Table 4). The change in vitamin D levels was statistically significant both in the placebo (11.8 ± 4.2 vs 15.2 ± 5.9 before and after treatment, respectively, p <0.001) and in the vitamin D supplementation (11.3 ± 3.6 vs 28.6 ± 10.3 before and after treatment, respectively, p <0.0001) groups. Table 4 demonstrates that vitamin D related parameters were similar at baseline between the study groups.

Table 5 displays same parameters after vitamin D (or placebo) supplementation. At the end of the study period 25(OH)D and 1,25(OH)2D levels increase and, PTH, VDBP levels decrease after vitamin D supplementation, yet, although LTL levels increase in both groups, the increase is more prominent in the placebo group (Table 5).

**Table 2. Demographic parameters in subjects.**

| Variables | Study group (n = 102) | Vitamin D group (n = 52) | Placebo group (n = 50) | p-value* |
|---|---|---|---|---|
| Age (years) | 58.4 ± 8.0 | 59.2 ± 8.6 | 57.7 ± 7.4 | 0.34 |
| Weight (kg) | 69.6 ± 9.1 | 68.8 ± 9.5 | 70.4 ± 8.6 | 0.58 |
| Height (cm) | 161.2 ± 5.5 | 161.4 ± 5.8 | 161.1 ± 5.2 | 0.76 |
| BMI (kg/m$^2$) | 26.7 ± 3 | 26.4 ± 3.2 | 27 ± 2.7 | 0.26 |
| Menopause age | 47.5 ± 4 | 47 ± 4.9 | 48.1 ± 4.3 | 0.21 |
| Smoking status | n (%) | n (%) | n (%) | 0.20 |
| *Smoker* | 26 (24.5) | 13 (23) | 13 (26) | |
| *Nonsmoker* | 77 (75.5) | 40 (77) | 37 (74) | |
| Exercise status | n (%) | n (%) | n (%) | 0.94 |
| *Moderate-exercisers* | 9 (8.8) | 5 (9.6) | 4 (8) | |
| *Mild-exercisers* | 19 (18.6) | 10 (19.2) | 9 (18) | |
| *Non-exercisers* | 74 (72.5) | 37 (71.2) | 37 (74) | |

Mean values ± standard deviations were shown when parameters were distributed normally. Median and (IQR) were shown when the parameters were in abnormal distribution.

*Significance between groups is shown as p-value <0.05.

**Table 3. Baseline biochemical characteristics of subjects.**

| Laboratory | Study group | Vitamin D group | Placebo group | p-value[*] |
|---|---|---|---|---|
| **Total cholesterol (mg/dl)** | 208.8±41.2 | 199.6±31.8 | 213.6±45.5 | 0.42 |
| **LDL-cholesterol (mg/dl)** | 115.1±35.9 | 107.3±30.5 | 119.3±38.8 | 0.45 |
| **HDL-cholesterol (mg/dl)** | 62.5±12.5 | 59.3±11.8 | 64.4±12.8 | 0.34 |
| **Triglyceride (mg/dl)** | 173.8±137.4 | 126 (180) | 117.5 (104.8) | 0.37 |
| **Fasting glucose (mg/dl)** | 102.2±22.2 | 94 (36.5) | 96.5 (26.9) | 0.84 |
| **Creatinine (mg/dl)** | 0.8±0.1 | 0.85 ± 0.2 | 0.9± 0.1 | 0.53 |
| **Alanine transaminase (ALT) (U/l)** | 15.8±6.2 | 15.0 (11) | 13.0 (5.0) | 0.29 |
| **Aspartate transaminase (AST) (U/l)** | 19,1±6.7 | 17.3 (3.3) | 17.6 (4.8) | 0.79 |

Mean values ± standard deviations were shown when parameters were distributed normally. Median and (IQR) were shown when the parameters were in abnormal distribution.

[*]Significance between groups is shown as p-value <0.05 and labeled as bold.

## Relative leukocyte telomere length

LTL was similar between supplementation and placebo groups before treatment. After the treatment, there was a statistically significant difference between the treatment and placebo groups (Table 6). Within the groups, there were significant seasonal changes in LTL (Table 6).

**Table 4. Baseline levels of variables in vitamin D supplementation and placebo groups.**

| Variables | Vitamin D group (n = 52) | Placebo group (n = 50) | p-value[*] |
|---|---|---|---|
| **25(OH)D (ng/ml)** | 11.2±3.6 | 11,8±4,2 | 0.534 |
| **1,25 (OH)$_2$D (pg/ml)** | 70.2±22.7 | 74.5±34.3 | 0.454 |
| **PTH (pg/ml)** | 47.5(29.6) | 54.4(33.9) | 0.084 |
| **LTL** | 6.0±1.4 | 6.1±1.3 | 0.751 |
| **VDBP levels (mg/l)** | 538.0±224.1 | 465.6±297.1 | 0.180 |

Mean values ± standard deviations were shown when parameters were distributed normally. Median and (IQR) were shown when the parameters were in abnormal distribution.

[*]Significance between groups is shown as p-value <0.05 and labeled as bold.

**Table 5. Change in variables after treatment in vitamin D supplementation and placebo groups.**

| Variables | Vitamin D group (n = 52) | Placebo group (n = 50) | p-value[*] |
|---|---|---|---|
| **25(OH)D (ng/ml)** | 28.6±10.3 | 15.2±5.9 | **<0.0001** |
| **1,25 (OH)$_2$D (pg/ml)** | 93.0±30.8 | 81.3±27.4 | **0.046** |
| **PTH (pg/ml)** | 29.9(26.8) | 41.1 (24.9) | **0.019** |
| **LTL** | 7.3±0.9 | 7.7±0.9 | **0.01** |
| **VDBP levels (mg/l)** | 433.7±198.1 | 352.9±170.1 | **0.034** |
| ***GC* expression change ($2^{-\Delta\Delta Ct}$)** | 0.58(0.42) | 0.9(0.9) | **0.012** |
| ***VDR* expression change ($2^{-\Delta\Delta Ct}$)** | 0.17(0.9) | 0.4 (1.5) | 0.18 |

Mean values ± standard deviations were shown when parameters were distributed normally. Median and (IQR) were shown when the parameters were in abnormal distribution.

[*]Significance between groups is shown as p-value <0.05 and labeled as bold.

**Table 6. Comparison of relative telomere length (ΔCt), in treatment and placebo groups according to the season of treatment.**

| Variables | Vitamin D group (n = 52) | Placebo group (n = 50) | p-value* |
|---|---|---|---|
| **LTL** | | | |
| **before treatment (summer)** | 5.29±1.06 (n = 25) | 5.67±1.16 (n = 39) | 0.20 |
| **after treatment (summer)** | 7.46±0.63 (n = 25) | 7.72±0.73 (n = 39) | 0.14 |
| **p-value** | <**0.0001** | <**0.0001** | |
| **before treatment (winter)** | 6.69±1.24 (n = 27) | 7.67± 0.47 (n = 11) | **0.001** |
| **after treatment (winter)** | 7.08±1.06 (n = 27) | 7.72±1.28 (n = 11) | 0.16 |
| **p-value** | 0.22 | 0.90 | |

Mean values ± standard deviations were shown as parameters were distributed normally.

*Significance between groups is shown as p-value <0.05 and labeled as bold.

LTL increased significantly in summer in both groups, yet no significant difference was noted in winter (Table 6).

### *VDR* and *GC* mRNA expressions and VDBP levels

The concentrations of VDBP protein in human plasma are normally maintained within a relatively narrow range (350–550 mg/l [6.25 to9.8 pmol] [26].

VDBP levels were similar at baseline (Table 4). After treatment, VDBP levels were higher in the treatment group compared to the placebo (Table 5).

When treatment vs placebo groups were compared, the decrease in VDBP levels and change in *GC (VDBP)* mRNA expression were statistically significant (Table 5).

The fold change ($2^{-\Delta\Delta Ct}$) in groups was compared as the change in *VDR* expression. The difference in mRNA expressions of *VDR* was not statistically significant in treatment vs placebo groups. When we compared *VDR* expression with smoking in the whole group, VDR expression did not change in smokers (p = 0.786), but significantly decreased in non-smokers (p = 0.007). *GC* (*VDBP*) expression did not change in smoking (p = 0.498) vs non-smoking (0.373) groups. Physical activity had no effect on *VDR* expression.

## Discussion

Vitamin D deficiency is a common risk factor for aging and age-related diseases [11, 27]. Telomere length is one of the potential protective mechanisms especially for age-related diseases including cancer, diabetes and cardiovascular diseases [28]. The short-term effects of Vitamin D supplementation on telomere length are largely unknown. We evaluated the short-term effects of vitamin D supplementation on LTL and on the expressions of vitamin D associated genes in postmenopausal women. We have chosen post-menopausal women to exclude the estrogen effects on vitamin D [29, 30]. In order to establish homogeneity of the groups, we selected a village in western Anatolia with ample seasonal sun exposure. The homogeneity of the group enables us to observe the effects of treatment better, minimizing the confounding effects.

At the end of the study period 25(OH)D and 1,25(OH)$_2$D levels increase and, PTH, VDBP levels decrease after vitamin D supplementation, yet, although LTL levels increase in both groups, the increase is more prominent in the placebo group. Given the study design we cannot make conclusions about the effects of seasonal changes on LTL. However, LTL significantly increases in both groups in summer months, suggesting that the relation between LTL and vitamin D levels is complex and can be affected by sun exposure and seasonal changes.

The study region experienced particularly hot summers in 2018–2019, which could have affected the study results [31].

The characteristics of the study location and population i.e. obesity, diet and socioeconomic factors can be other confounding factors [32, 33]. Our findings support prior studies. The seasonal changes may cause variability in vitamin D levels among subjects with high BMI. 1,25 $(OH)_2D$ levels are lower in obese and elderly persons [32].

The study results suggest that telomere length can be a potential compensatory mechanism in persistent vitamin D deficient states. The active form of vitamin D is $1,25(OH)_2D$. $1,25$ $(OH)_2D$ levels are increased after vitamin D supplementation compared to the placebo with borderline statistical significance. Prior study reports that $1,25(OH)_2D$ levels remain constant throughout the year. Age and BMI can affect $1,25(OH)_2D$ levels [33]. Our study subjects were overweight and postmenopausal which can affect the $1,25(OH)_2D$ levels.

PTH levels decreased with vitamin D supplementation. PTH plays a role in the regulation of vitamin D metabolism and stimulates the formation of $1,25(OH)_2D$ in the kidneys. There is an inverse association between PTH levels and vitamin D deficiency and up to a 20% decrease is expected with vitamin D supplementation in normal weight individuals [34, 35]. However, the dose of vitamin D supplementation to suppress PTH levels may differ in overweight and obese adults [36].

Vitamin D is reported to have an effect on telomere length due to its role in cell proliferation, aging and apoptosis [16, 37, 38]. The molecular mechanisms of interaction between Vitamin D and telomere length are largely unknown. Telomere shortening is directly proportional to age and oxidative stress. Vitamin D replacement reduces nuclear factor–kB activity, increases anti-inflammatory IL10 levels, decreases pro-inflammatory interleukins, therefore acts on cell survival [39]. However, although 25(OH)D is associated with telomere levels, it is not known whether vitamin D replacement has any effect on LTL. There are conflicting reports about vitamin D levels and LTL. A positive correlation between 25(OH)D levels with LTL is reported [37, 38]. Yet, these studies are cross sectional and did not examine the seasonal changes and/or vitamin D supplementation on LTL levels.

Our observations suggest that the relation between Vitamin D and LTL is not linear and can be affected by several confounding factors such as the populational, regional and seasonal changes. Prospective studies are needed to monitor the LTL in relation to vitamin D levels. Our observations can add to the literature as it is a placebo-controlled study among postmenopausal women showing the short-term effect of Vitamin D supplementation on LTL together with vitamin D associated gene expressions.

A prior study that demonstrates the effects of vitamin D supplementation in Afro-American subjects associated the effects of vitamin D supplementation with telomerase activity and telomerase activity increases with vitamin D supplementation. However, telomere length can also be preserved independent of the activity of the telomerase enzyme [40]. Julin et al. reported that no correlation exists between 25(OH)D, $1,25(OH)_2D$ and LTL in men [41]. Yang et al. demonstrate that vitamin D supplementation for 12 months increases the telomere length in elderly subjects with mild cognitive impairment (MCI) who received oral vitamin $D_3$ daily for 12-month period [16].

VDBP is the primary vitamin D carrier; binding almost 90% of circulating vitamin D and the unbound (or bound to albumin) is the bioavailable part. Menopause represents an important transition in vitamin D requirement [42]. VDBP and 25(OH)D levels are significantly higher in premenopausal women than postmenopausal women and estradiol levels are correlated with VDBP and 25(OH)D [43]. Individual and ethnic differences can affect VDBP level and attachment capacity. In our study, we observed a that VDBP levels decrease and 25(OH)D levels increase with treatment in both placebo and supplementation groups. The change in

mRNA expression of *GC(VDBP)* is significant when treatment and placebo groups were compared. The placebo group showed a more prominent increase in expression compared to the treatment group. The gene expression and protein levels may be discordant according to differences in regulation at the protein or transcriptional level. Protein degradation may be slowed or accelerated according to changing transcript levels [44]. VDR mediates 1,25 $(OH)_2D$ and its functions. *VDR* expression remained stable independent of treatment.

In our study, we observe the short-term effects of vitamin D supplementation. The homogeneity of the group enables us to observe the effects of treatment better, minimizing the confounding effects. The limitations of our study include the small size of the study. In Turkey almost 50% of postmenopausal women have obesity and metabolic syndrome [45]. The study included subjects with high BMI which can limit the generalizability of results to the lean population [46]. We applied the same dose of vitamin D to all participants but metabolism of vitamin D varies between individuals, therefore dose adjustments can yield different results in overweight individuals. We did not have detailed dietary and physical activity scores from the participants. Socioeconomic factors, household income, and ecological and environmental factors can all confound the study findings Even though we mention the seasonal effects on telomere length, the design of the study precludes us to make any conclusive statements on seasonal changes. An appropriate design would have been to include the season of recruitment in the stratification matrix at the time of randomization to exclude the confounding factors. Unfortunately, due to limited funding for a precise period of time, the speed of enrollment, it remained to be underpowered to investigate the effects of seasons on telomere length. We cannot claim that the findings apply to global populations, i.e. unlike western European populations. Due to the climate, the location of the study receives ample amount of sunlight even in the winter months. The results can be different in a different location with limited sun exposure in the winter months.

## Conclusions and future directives

LTL displays dynamic changes in postmenopausal women with vitamin D deficiency. Yet the interaction appears more complex than a linear relationship possibly attributable to persistent vitamin D deficiency, and other confounding factors. This study provides a short term evaluation of the effects of vitamin D supplementation on LTL and vitamin D related biomarkers in postmenopausal women with vitamin D deficiency. The results indicate that vitamin D supplementation and seasonal changes can insert bidirectional effects on vitamin D related parameters including LTL and VDBP. Large prospective studies in diverse populations are needed to understand the effect of vitamin D levels and seasonal changes on LTL and vitamin D related parameters.

## Supporting information

**S1 Dataset.**
(ZIP)

## Author Contributions

**Conceptualization:** Deniz Agirbasli, Minenur Kalyoncu, Meltem Muftuoglu, Fehime Benli Aksungar, Mehmet Agirbasli.

**Data curation:** Deniz Agirbasli.

**Formal analysis:** Deniz Agirbasli, Minenur Kalyoncu, Fehime Benli Aksungar, Mehmet Agirbasli.

**Funding acquisition:** Deniz Agirbasli.

**Investigation:** Deniz Agirbasli, Minenur Kalyoncu.

**Methodology:** Deniz Agirbasli, Meltem Muftuoglu, Fehime Benli Aksungar, Mehmet Agirbasli.

**Project administration:** Deniz Agirbasli.

**Resources:** Deniz Agirbasli, Meltem Muftuoglu.

**Software:** Deniz Agirbasli, Fehime Benli Aksungar, Mehmet Agirbasli.

**Supervision:** Deniz Agirbasli, Meltem Muftuoglu, Mehmet Agirbasli.

**Validation:** Deniz Agirbasli, Fehime Benli Aksungar, Mehmet Agirbasli.

**Visualization:** Deniz Agirbasli.

**Writing – original draft:** Deniz Agirbasli.

**Writing – review & editing:** Deniz Agirbasli, Minenur Kalyoncu, Meltem Muftuoglu, Fehime Benli Aksungar, Mehmet Agirbasli.

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
