## [Decision Letter · Decision Letter 0]

1 Dec 2021

PONE-D-21-17303

Leukocyte telomere length as a compensatory mechanism in vitamin D metabolism

PLOS ONE

Dear Dr. Agirbasli,

Thank you for submitting your manuscript to PLOS ONE. After careful consideration, we feel that it has merit but does not fully meet PLOS ONE’s publication criteria as it currently stands. Therefore, we invite you to submit a revised version of the manuscript that addresses the points raised during the review process.

ACADEMIC EDITOR: The study and manuscript have merit, but need a better description of sample size determination and other improvements (Introduction etc.). Please adhere closely and address the concerns of the reviewers.

We look forward to receiving your revised manuscript.

Kind regards,

Gabriele Saretzki, PhD

Academic Editor

PLOS ONE

Journal Requirements:

Additional Editor Comments:

The pilot study seems interesting and of good quality.

Reviewers' comments:

Reviewer's Responses to Questions

**Comments to the Author**

1. Is the manuscript technically sound, and do the data support the conclusions?

Reviewer #1: Yes

Reviewer #2: No

2. Has the statistical analysis been performed appropriately and rigorously? 

Reviewer #1: Yes

Reviewer #2: No

3. Have the authors made all data underlying the findings in their manuscript fully available?

Reviewer #1: Yes

Reviewer #2: No

4. Is the manuscript presented in an intelligible fashion and written in standard English?

Reviewer #1: Yes

Reviewer #2: Yes

5. Review Comments to the Author

Reviewer #1: The topic of this paper is of scientific interest. The paper by Agirbasli et al. deals with leucocyte telomere length (LTL) and vitamin D, especially the effect of vitamin D supplementation on LTL in postmenopausal women having vitamin D deficiency. This short-term placebo-controlled study investigates vitamin D supplementation and seasonal changes of vitamin D related parameters. The following parameters were measured before and after treatment: 25(OH)D, 1,25(OH)2D, PTH, VDBP, VDR, and LTL.

It has been shown that serum vitamin D levels increase by its supplementation while LTL is influenced by sun exposure and seasonal periods. In the summer time LTL is higher than in winter season. VDBP decreases with vitamin D supplementation. The findings demonstrate that vitamin D levels are related to aging and to genomic stability. As vitamin D reduces inflammation it is related to telomere length and in that way to genomic integrity. The findings underline that vitamin D plays an important role in LTL changes by vitamin D supplementation but also by variation in sun exposure, summer vs. winter time.

This short-term study describes effects of vitamin D supplementation on LTL and vitamin D related biomarkers in postmenopausal women with vitamin D deficiency. The findings indicate that vitamin D supplementation and seasonal changes have bidirectional effects on vitamin D related parameters including LTL and VDBP. The paper is well written. The tables are informative and demonstrate the findings very well.

Reviewer #2: PONE-D-21-17303

Leukocyte telomere length as a compensatory mechanism in vitamin D metabolism

The role of vitamin D in preventing genome damage has been associated with a multitude of disease conditions. Telomeres play a major role is capping chromosome ends to prevent DNA shortening, thus playing a vital role in preserving genome integrity. There have been some evidence to suggest that vitamin D may prevent genome damage and this effect may also be apparent for maintaining telomere length (TL). The authors have conducted a clinical trial to determine if vitamin D supplementation was associated with TL. They also attempted to examine the seasonal effect of this supplementation. The authors have recognised the role of latitude on vitamin D levels and thus selected participants from the one region to minimise this effect. The topic is topical however I have some concerns regarding the study design and interpretation of results.

Major:

1. The authors have not included a sample size calculation. Their primary aim was to determine if there was an association between D-supplementation and TL. A secondary aim was to additionally examine a seasonal effect which they would have been under powered to examine with a sample size of 102. An appropriate design would have been to include season of recruitment in the stratification matrix at time of randomisation. Instead, their results have been stratified by season at time of analysis which is not appropriate, with no analysis on whole group analysis. As a consequence the placebo group only contained a fifth of the recruited participants (n=11) and would have been way under powered to generate anything meaningful. They should only be publishing data on whole group analysis.

2. Additionally the authors should not be comparing inter group analysis as this is not a phase 3 trial. They should be reporting results from intra group analysis.

3. The authors have not described how the blinding and randomisation was carried out. They claim that the participants and laboratory staff were blinded to the study arm, where the intervention was Devit-3 Deva oral solution and placebo was sunflower oil. Were these ampules/bottles packaged to look identical to ensure that the participants were truly blinded. If so, they need to describe this process.

4. The authors have not described the vitamin D metabolite used for intervention i.e. is it cholecalciferol or ergocalciferol?

Minor:

1. Authors need to include I their Introduction what the evidence is on TL status in postmenopausal women, else what is the purpose of studying this relationship in this group.

2. Line 86: they have described using ‘healthy” postmenopausal women. Can they please describe definition for ‘healthy”.

6. PLOS authors have the option to publish the peer review history of their article (what does this mean?). If published, this will include your full peer review and any attached files.

Reviewer #1: No

Reviewer #2: No

---

## [Author Response · Author response to Decision Letter 0]

6 Jan 2022

Dear Editor,

We greatly appreciate the time, effort, and substantive comments of the reviewers. We have incorporated changes in accordance with reviewer suggestions, or provided rationale for not incorporating the respective suggestion. We would like to thank the reviewers for helping us to improve the clarity and substance of our paper. The reviewers’ valuable comments helped us refine the manuscript in preparation for publication. The reviewers’ comments have been included below for reference with a subsequent response.

Reviewer #1:

 The topic of this paper is of scientific interest. The paper by Agirbasli et al. deals with leucocyte telomere length (LTL) and vitamin D, especially the effect of vitamin D supplementation on LTL in postmenopausal women having vitamin D deficiency. This short-term placebo-controlled study investigates vitamin D supplementation and seasonal changes of vitamin D related parameters. The following parameters were measured before and after treatment: 25(OH)D, 1,25(OH)2D, PTH, VDBP, VDR, and LTL.

It has been shown that serum vitamin D levels increase by its supplementation while LTL is influenced by sun exposure and seasonal periods. In the summer time LTL is higher than in winter season. VDBP decreases with vitamin D supplementation. The findings demonstrate that vitamin D levels are related to aging and to genomic stability. As vitamin D reduces inflammation it is related to telomere length and in that way to genomic integrity. The findings underline that vitamin D plays an important role in LTL changes by vitamin D supplementation but also by variation in sun exposure, summer vs. winter time.

This short-term study describes effects of vitamin D supplementation on LTL and vitamin D related biomarkers in postmenopausal women with vitamin D deficiency. The findings indicate that vitamin D supplementation and seasonal changes have bidirectional effects on vitamin D related parameters including LTL and VDBP. The paper is well written. The tables are informative and demonstrate the findings very well.

We appreciate the kind comments of the reviewer.

Reviewer #2: 

PONE-D-21-17303

Leukocyte telomere length as a compensatory mechanism in vitamin D metabolism

The role of vitamin D in preventing genome damage has been associated with a multitude of disease conditions. Telomeres play a major role is capping chromosome ends to prevent DNA shortening, thus playing a vital role in preserving genome integrity. There have been some evidence to suggest that vitamin D may prevent genome damage and this effect may also be apparent for maintaining telomere length (TL). The authors have conducted a clinical trial to determine if vitamin D supplementation was associated with TL. They also attempted to examine the seasonal effect of this supplementation. The authors have recognised the role of latitude on vitamin D levels and thus selected participants from the one region to minimise this effect. The topic is topical however I have some concerns regarding the study design and interpretation of results.

Major:

1. The authors have not included a sample size calculation. 

We agree with the reviewer. We added the power analysis to the methods section.

‘When the type 1 error is set as 5% with study power 80%, with a hypothetical difference of 10% between the telomere length of the study groups, the minimum sample size is calculated as 38 for each group.’ (page 6, lines 125-127).

Their primary aim was to determine if there was an association between D-supplementation and TL. A secondary aim was to additionally examine a seasonal effect which they would have been under powered to examine with a sample size of 102. An appropriate design would have been to include season of recruitment in the stratification matrix at time of randomisation. Instead, their results have been stratified by season at time of analysis which is not appropriate, with no analysis on whole group analysis. As a consequence the placebo group only contained a fifth of the recruited participants (n=11) and would have been way under powered to generate anything meaningful. They should only be publishing data on whole group analysis.

We agree with insightful comments of the reviewer. To provide the whole group analysis comparing the groups vitamin D supplementation with placebo, we revised the table 4 and added table 5 as suggested by the reviewer and deleted results associated to seasonal changes. We rewrote the discussion after excluding the statistics on seasonal effects and added the limitations to the discussion. We gave the results of the telomere length both in the placebo and treatment groups before and after 2 months of treatment and 1 month waiting period. This remains to be the main finding of the study. Despite an increase in vitamin D levels and decrease in PTH levels in the treatment group compared to the placebo arm, the telomere length was longer in the placebo group compared to the active treatment arm at the end of study period. We agree with the reviewer that an appropriate design would definitely have been to include the season of recruitment in the stratification matrix at time of randomization to exclude the confounding factors. Unfortunately, due to limited funding for a precise period of time, and the speed of enrollment, study could not be designed as suggested by the reviewer, and therefore remained to be underpowered to investigate the effects of seasons on telomere length. We acknowledge the limitations as indicated by the reviewer. The subgroup analysis remains in telomere length with intra group analysis in table 6 as suggested by the reviewer. We omitted previous table 6 showing the seasonal effects of vitamin D related parameters as suggested by the reviewer. We will be willing to shorten the results and the tables further per reviewer suggestions. 

2. Additionally the authors should not be comparing inter group analysis as this is not a phase 3 trial. They should be reporting results from intra group analysis.

We agree with the reviewer. We revised tables 4 and 5 and omitted table 6 as suggested by the reviewer. We added the limitations to the discussion. 

3. The authors have not described how the blinding and randomisation was carried out. They claim that the participants and laboratory staff were blinded to the study arm, where the intervention was Devit-3 Deva oral solution and placebo was sunflower oil. Were these ampules/bottles packaged to look identical to ensure that the participants were truly blinded. If so, they need to describe this process.

“Vitamin D supplementation bottles were packaged to look identical to ensure that the participants were truly blinded.” The explanation is added to methods section (page 6, lines 139-140). 

4. The authors have not described the vitamin D metabolite used for intervention i.e. is it cholecalciferol or ergocalciferol? 

The vitamin D metabolite used for intervention is cholecalciferol. The information is added to the methods section (page 6, line 134).

Minor:

1. Authors need to include I their Introduction what the evidence is on TL status in postmenopausal women, else what is the purpose of studying this relationship in this group.

We thank the reviewer for the careful consideration of the manuscript. According to the comments of the reviewer, a paragraph is added to Introduction section explaining the LTL status in postmenopausal women and the purpose of studying this relationship in this group with associated references (page 4, lines 78-84).

“There are numerous studies investigating the LTL in postmenopausal women. Although ethnicity does not play a role in LTL, age and gender are among the determinants of LTL [13,14]. Decreased estrogen levels after menopause, a pivotal factor in the biology of aging, were positively associated with LTL. [13]. Studies also indicate from the premenopausal period through the perimenopausal period to the postmenopausal period there is gradual attrition in LTL which than turned out to be more stable after the postmenopausal period [15].

Shin YA, Lee KY. Low estrogen levels and obesity are associated with shorter telomere lengths in pre- and postmenopausal women. J Exerc Rehabil. 2016 Jun 30;12(3):238-46. 

Jones HJ, Janson SL, Lee KA. Leukocyte Telomere Length in Postmenopausal Women. J Obstet Gynecol Neonatal Nurs. 2017;46(4):567-575. 

Dalgård C, Benetos A, Verhulst S, Labat C, Kark JD, Christensen K, Kimura M, Kyvik KO, Aviv A. Leukocyte telomere length dynamics in women and men: menopause vs age effects. Int J Epidemiol. 2015 Oct;44(5):1688-95.

2. Line 86: they have described using ‘healthy” postmenopausal women. Can they please describe definition for ‘healthy”.

 We added the explanation under the title “Study Population” in Methods section (page 5, lines 96-100). “According to the definition of World Health Organization (WHO); health is a state of complete physical, mental, and social well-being. The study population was defined as healthy, indicating that they are independent any known diseases without any prior discerning medical history in accordance with the definition of WHO.”

---

## [Decision Letter · Decision Letter 1]

9 Feb 2022

Leukocyte telomere length as a compensatory mechanism in vitamin D metabolism

PONE-D-21-17303R1

Dear Dr. Agirbasli,

We’re pleased to inform you that your manuscript has been judged scientifically suitable for publication and will be formally accepted for publication once it meets all outstanding technical requirements.

Kind regards,

Gabriele Saretzki, PhD

Academic Editor

PLOS ONE

Additional Editor Comments (optional):

All remaining issues have been addressed.

Reviewers' comments:

Reviewer's Responses to Questions

**Comments to the Author**

1. If the authors have adequately addressed your comments raised in a previous round of review and you feel that this manuscript is now acceptable for publication, you may indicate that here to bypass the “Comments to the Author” section, enter your conflict of interest statement in the “Confidential to Editor” section, and submit your "Accept" recommendation.

Reviewer #2: All comments have been addressed

2. Is the manuscript technically sound, and do the data support the conclusions?

Reviewer #2: Yes

3. Has the statistical analysis been performed appropriately and rigorously? 

Reviewer #2: Yes

4. Have the authors made all data underlying the findings in their manuscript fully available?

Reviewer #2: Yes

5. Is the manuscript presented in an intelligible fashion and written in standard English?

Reviewer #2: No

6. Review Comments to the Author

Reviewer #2: PONE-D-21-17303

Leukocyte telomere length as a compensatory mechanism in vitamin D metabolism

The role of vitamin D in preventing genome damage has been associated with a multitude of disease conditions. Telomeres play a major role is capping chromosome ends to prevent DNA shortening, thus playing a vital role in preserving genome integrity. There have been some evidence to suggest that vitamin D may prevent genome damage and this effect may also be apparent for maintaining telomere length (TL). The authors have conducted a clinical trial to determine if vitamin D supplementation was associated with TL. They also attempted to examine the seasonal effect of this supplementation. The authors have recognised the role of latitude on vitamin D levels and thus selected participants from the one region to minimise this effect. These findings will add to our growing knowledge on the relationship between vitaD and genome damage in cancer risk.

7. PLOS authors have the option to publish the peer review history of their article (what does this mean?). If published, this will include your full peer review and any attached files.

Reviewer #2: No

---

## [Editor Report · Acceptance letter]

14 Feb 2022

PONE-D-21-17303R1 

Leukocyte telomere length as a compensatory mechanism in vitamin D metabolism 

Dear Dr. Agirbasli:

I'm pleased to inform you that your manuscript has been deemed suitable for publication in PLOS ONE. Congratulations! Your manuscript is now with our production department. 

Kind regards, 

on behalf of

Dr. Gabriele Saretzki 

Academic Editor

PLOS ONE